# The Parable of Wise and Foolish Builders in Yishen Lun and Rabbinic Literature

David Tam [1,2]

1    Institute for Ethics and Religious Studies, Tsinghua University, Beijing 100190, China; dtwtam@hotmail.com
2    Institute of Sino-Christian Studies, Hong Kong, China

**Abstract:** The paper undertakes a comparative analysis of the parable of the Wise and Foolish Builders as presented in three distinct sources: the seventh-century Dunhuang manuscript *Yishen Lun* (Discourse on God), the sixth-century rabbinic text *Avot D'Rabbi Nathan*, and the Gospels (Matthew and Luke) of the Christian Bible. It explores the imagery used, piety taught, and worldviews conveyed in these renditions, concluding that the version in *Yishen Lun* shares a closer resemblance with the one in rabbinic literature than with the Gospels. This discovery, in conjunction with previously published findings by the author, challenges the conventional classification of *Yishen Lun* as an "Aluoben document" (or a *Jingjiao* document, for that matter), underscoring the need for further research and inquiry.

**Keywords:** *Yishen Lun* (Discourse on God); the "Wise and Foolish Builders" parable; *Avot D'Rabbi Nathan*; Chinese Christianity; *Jingjiao* study

## 1. Introduction

In his 1918 article titled "*Keikyo Kyokten Isshinron Kaisetsu* 景教經典一神論解說 (An Explication of the *Jingjiao* Document *Yishen Lun*)" (Haneda 1918), Toru Haneda introduced the Dunhuang manuscript *Yishen Lun* (Discourse on God[1]) and identified it as a Christian document because of the "Sermon on the Mount" (a partial paraphrase) found in it. In subsequent papers from 1923 and 1931, Haneda (1923, 1931) dated the document to the year 641 CE[2] and published its text.[3]

*Yishen Lun* (YSL) can be divided into two parts:[4] Part I (lines 1–205) is a doctrinal treatise covering topics such as God, All Things, humanity, piety, two worlds (*citianxia-bitianxia*, 此天下/彼天下 or this world and that world), demons and evil. Part II (lines 208–404) provides an account of the teachings, trial, crucifixion, and resurrection of Jesus, along with a brief history from Pentecost to the time of the author. It calls on Christians to adhere to Christ's words and non-Christians to embrace the faith. The "Sermon on the Mount", as identified by Haneda, is found in the opening lines (208–249) of Part II and commences with the proclamation "Shizun yue 世尊曰 (The Lord[5] says)".

In the Gospels, the Sermon on the Mount, in either the Matthean or Lukan form, concludes with "The Wise and Foolish Builders" parable (Builders/Gospels), which exhorts people to act on Jesus' teachings rather than merely hearing them. This parable is an essential component of the Sermon on the Mount, serving as a powerful conclusion. As George Strecker puts it, "it places an impressive capstone on the structure of the Sermon". (Strecker 1988, p. 169). Given its importance, it is odd that in YSL, the parable does not appear at the end of its "Sermon on the Mount" (208–249), but at the end of a different passage, namely, the *citianxia-bitianxia* discourse (108–159) in Part I. This section is not attributed to Shizun and imparts a distinct lesson. Meanwhile, there exists another version of the "Wise and Foolish Builders" parable in Jewish rabbinic literature, specifically in the sixth-century *midrashic* work *Avot D'Rabbi Nathan* (The Fathers According to Rabbi Nathan), which is an extended commentary and amplification on the third-century *mishnaic* tractate *Pirkei*

*Avot* (Chapters of the Fathers). What does this version (Builders/Rabbinic) convey and teach, and how does it compare with the version in the *citianxia-bitianxia* passage of YSL (Builders/YSL)? Between Builders/Gospels and Builders/Rabbinic, with which one does Builders/YSL share a closer resemblance? These are the central questions addressed in this paper. In concluding his 2009 paper comparing the Gospel and rabbinic versions of the parable, Eric Ottenheijm remarks: "Often parallels have been considered problematic with regard to 'who influenced whom'. Our task was, however, to compare the discourses the parallels serve. From this perspective, some conclusions may be drawn." (Ottenheijm 2009, pp. 60–61). We share a similar focus in this paper.

Three common elements surface when scrutinizing the three versions of the parable in conjunction with their respective contextual discourses: the imagery of the parable, the two forms of piety, and a worldview consisting of two worlds. These three aspects provide the groundwork for our study and comparison, guiding the structure of the overview (Sections 2–4), the comparative analysis of the three versions (Section 5), and some concluding remarks (Section 6).

A couple of preliminary considerations warrant attention regarding the scope of this study. As mentioned, the objective of this paper is to assess whether Builders/YSL aligns more closely with Builders/Gospels or Builders/Rabbinic, and, as such, it does not directly involve a comparison between the latter two. We note only that the relationship between the Gospel and rabbinic versions of the parable has long been studied by scholars (e.g., Snodgrass 2008; Ottenheijm 2009; Jones 2012, etc.). On the question of whether Jesus borrowed the parable from the rabbis or vice versa, a widely shared view is that the rabbinic version, dating to about 500 CE and attributed to a second-century rabbi, might be based on earlier tradition, but it is challenging to argue for direct borrowing in either direction. The two versions are in fact not very similar, and their logic is straightforward enough that they might have emerged independently (Snodgrass 2008, p. 334).

Second, considering the presence of Buddhist vocabulary and phraseology in YSL,[6] one might wonder if Builders/YSL could have a Buddhist origin. However, *Jingjiao* scholars have cautioned against interpreting YSL (or other "*Jingjiao* texts" for that matter) with a Buddhist bias because of the language. Chen Huayu conjectures that the Buddhist language was there because the Christians in Tang China had relied on local literati to prepare their texts, and the only religious language these hired hands knew was that of the Buddhist or Taoist tradition (Chen 2015, p. 52). Huang Xianian contends that, while certain Buddhist elements may be present in Tang Christian texts ("*Jingjiao* documents"), the core Christian doctrines and perspectives within these texts remain intact. Therefore, any suggestions of potential Buddhist influences should be approached with careful consideration (Huang 2000, p. 90). More pertinent to this study is that while a house-and-foundation metaphor can also be found in Buddhist texts, a complete parable contrasting the success and failure of two different building methods tested by violent natural events would likely be rare, if not entirely absent, in Buddhist texts. Hence, a comparison with a Buddhist source is not within the scope of this paper.

## 2. The Parable in the *Citianxia-Bitianxia* Discourse of YSL

The following discussion involves an exegesis of YSL, specifically centered on the *citianxia-bitianxia* section. It is important to note from the outset that a consensus regarding the interpretation of YSL (and "*Jingjiao* documents" in general) is often lacking, particularly concerning the more difficult and opaque segments. The challenges of interpreting YSL have long been acknowledged by scholars. Haneda observed, "However, this translation [YSL] seems to be a kind of colloquial expression from the Tang Dynasty, making it very difficult to understand" (Haneda 1918, p. 236). Jiang Wenhan described the text as strange and difficult to understand (Jiang 1982, p. 59), while Nicolini-Zani characterized it as "difficult", "obscure", and "problematic" (Nicolini-Zani 2022, pp. 151, 194, 234, 252, etc.). In the view of this author, the optimal approach involves gaining a thorough familiarity with the text through repeated readings and structural analysis, consulting rel-

evant sources such as other Dunhuang and Turfan texts,[7] and referencing interpretations provided by fellow scholars. Within this framework, the author aims to present his own interpretations in the subsequent discussion, with the rationale for these interpretations provided in the footnotes, wherever possible.

Generally speaking, the *citianxia-bitianxia* discourse (108–159) in Part I discusses a rewarding system of human existence in two worlds. It says that just as our body in *citianxia* (this world) is prepared during our time in the mother's womb, our soul's state of being in *bitianxia* (the world-to-come) is determined by our deeds in *citianxia*. The text underlines the significance of performing good works in *citianxia*, drawing distinctions between a form of piety without good works (i.e., worship, study and rules observance only), and another with good works of almsgiving. This ethical teaching, based on a profound theory of the soul and worldview (91–107), finds expression through the parable in lines 146–156.

*2.1. Imagery*

146 唯事一神天尊，[8] 礼拜一神，一取一神進止[9]。不

You are to love and serve the only one God, the Lord. Worship God and adhere solely to God's commands.

147 是[10]此意知功德，不是餘處功德。[11] 此處功德，

[However,] this is the good works of the mind, not all the good works. The good works referred to here

148 不是功德處。[12] 喻如人作舍，預前作基腳，先須

does not encapsulate the fundamental nature of good works. For example, someone builds a house by first laying down a foundation, making it

149 牢固安置。[13]若基腳不牢固，舍即不成。喻如欲

firm and secure. If the foundation is not solid, the house is not complete. This is to say that

150 作功德，先脩行，具戒俻足，[14] 亦須知一神安置。

when we do good works, we self-cultivate, observe all the rules, and be cognizant of the providence of God.

151 人皆須礼拜，須領一神恩，然後更別作[15]功德。

[Indeed,] everyone should worship God and receive His grace, but should do all the other good works as well.

152 此是言語讚歎功德，亦不是餘功德。亦須知，喻

[Otherwise,] this is good works of verbal praise and adoration, not the rest of all good works. One should know, with respect to

153 如說言，須作好善意，智裏天尊。何誰別在功

speech, one ought to be good-intentioned, with the mind on the Lord. If one is not

154 德處不勤心時，如似人無意智，欲作舍，基腳

engaged in good works diligently, he or she would be like a foolish person building a house with a foundation

155 不著地，被風懸吹[16]將去。如舍腳牢，風亦不能懸

not properly set in the ground, causing the house to be blown away by the swirling wind. If the foundation is firmly built, the wind cannot

156 吹得。…

blow it away …

There are five movements in these lines. First (146–148a), it exhorts people to love, worship, and obey the only one God, while also underscoring that this alone does not encompass all good works, nor does it address the central meaning of good works. Second (148b–149), it introduces the parable, saying that when constructing a house, laying a solid foundation is crucial, as the project would otherwise fail. Third (150–153a), it admonishes those who are pious only in the sense that they sing and praise God, study the scriptures and observe the rules, cautioning that they may not fully understand the true will of God, as they neglect other good works. Fourth (153b–155a), it compares those who ignore doing the other good works to a foolish person building a house without laying a solid foundation, leading to the house being easily blown away by the wind. Fifth (155b–156a), it reassures that if the foundation is solid, the house will remain secure.

### 2.2. Two Forms of Piety

The first form of piety is depicted in line 146, and it involves loving, worshipping, and obeying the only one God. The subsequent lines emphasize that while this piety is correct and important, if it is not supplemented by other good works, it is not secure at the foundation and it will fail. A complete sense of piety is outlined in lines 130–140:

130b–131 一切功德，須此處作，不是彼處作。

For all the good works, they should be done here, not over there.

131–132 莫跪拜鬼。此處作功德，不是彼處。

Do not worship the ghosts. Do the good works here, not over there.

132–133 一神處分[17]莫違。願此處得作，彼處不得作。

Do not go against the order (will) of God. It is wished that it (the will of God) be undertaken here, for it cannot be over there.

133–134 喻如作功德，先須此處作，不是彼處。

In terms of doing good works, they should be done here beforehand, not over there.

134–135 布施與他物功德，此處施得，彼處雖施亦不得。

For the good works of giving alms, they can be accomplished here, not over there even if one tries.

135–136 發心[18]須寬大，不得窄小，即得作寬，此處得作，彼處作不得。

The resolve should be inclusive in perspective, should not be narrow, so that [the good works] can benefit widely. This can be done here, not over there.

136–138 以此思量，毒心、惡意、怨酬、增[憎]嫉物，須除却，此處除可得，彼處除不可得。

With this in mind, any malice, venom, revenge and jealousy should be eliminated. They can be eliminated here, not over there.

138–139 身心淨潔，恭敬礼拜，不犯戒行，此處作得，彼處作不得。

Keep the body and mind pure and clean. Worship in reverence and commit no transgressions. These can be done here, not over there.

139–140a 至心礼拜天尊，一切罪業皆得除免。此處礼得，彼處礼不得。

Worship the Lord in utmost sincerity, so that all your sins will be forgiven. It is possible to worship here, but not over there.

In this description, the wholistic piety, on the negating side, includes rejecting the worship of demons and purging one's heart of malice, venom, revenge, or jealousy towards others, and on the confirming side, loving, serving, and worshipping the only one God (146: 唯事一神天尊，恭敬礼拜，至心礼拜), adhering solely to His commands, maintaining cleanliness in mind and body, and providing material assistance to those in need.

This passage possesses a rhythmic and poetic quality, characterized by the refrain "doing it here, not over there" placed at the end of each sentence, juxtaposing the terms "here" (*cichu* 此處) and "there" (*bichu* 彼處) repeatedly. These terms are abbreviations of *citianxia* and *bitianxia*, used perhaps for metrical considerations. The entire passage appears to be deliberately crafted for easier recitation and memorization, both for individuals and the community. This special treatment indicates the passage's central significance within this faith community.

### 2.3. Two Worlds

Builders/YSL, therefore, is a metaphor built on a worldview consisting of two worlds: *citianxia* and *bitianxia*. What happens in *citianxia* affects *bitianxia*. Thus:

121b–122 万物彼天下須，[此]天下須在前。此間須作，分明宣說。

All Things is required in *bitianxia*. *[Ci]tianxia* therefore should come first. What needs to be done here has now been clearly manifested.

129b–130 如彼天下須者，此間合作[19]，此間若不合作，至彼處亦不能作。

For what are mandatory in *bitianxia*, it is appropriate to have them done here. If they are not suitable to be done here, they are not over there either.

For this reason, one's condition in *bitianxia* is contingent upon his or her behavior and life choices in *citianxia*. While the changes may only manifest in the future, they are determined by what has transpired beforehand (116: 更在後，亦如在先作). The premise is that genuine good deeds in this life will lead to rewards in the next (119–120: 為如此生能脩善種，果報彼天下). The merits of these good deeds are subject to God's judgment (156: 如功德無天尊證，即不成就), and as for what they are, the verses in 130–140 have provided a sketch.

Although *bitianxia* is experienced by a person after *citianxia*, it does not imply that it comes into existence only after the conclusion of *citianxia*. The two worlds are concurrent, and one transitions to *bitianxia* upon departing from *citianxia*. This can be observed from 140–141, which states that a person proceeds to *bitianxia* after leaving *citianxia*, and from 142–144, which suggests that God (or the Power of God) proceeds to *bitianxia* after having established *citianxia*.

140–141 若有此天下去人，於此處種果報得具足。

For people who have gone (there) from *citianxia*, they will receive all the rewards for what they have sown here.

142–144 一神自聖化神自聖化，[20]神力作，在先安置天下，然後彼天下去。

God is self-sanctified. The power of God has acted, and having provided for *citianxia*, He proceeds to *bitianxia*.

The concept of *bitianxia* in YSL is, therefore, more of a cosmological worldview than an eschatological one, and this is evident in the frequent juxtaposition of the paired term *citianxia* and *bitianxia* (or their variants).

### 3. The Parable in *Avot D'Rabbi Nathan* of Rabbinic Literature

The original *mishnaic* tractate *Pirkei Avot* delivered its teachings through aphorisms attributed to sages around 250 CE. Later, possibly around 500 CE, *Avot D'Rabbi Nathan* expanded on this tractate by adding numerous narratives, presenting a significant number of stories and parables, (Neusner 1997, p. xli) and one of these elaborations and additions is the parable of the two builders, attributed to Elisha b. Abuyah, a second-century rabbi. The parable is found in Paras A–D, Section I, Chapter XXIV of Neusner's edition of *Avot D'Rabbi Nathan*:

### 3.1. Imagery

A. Elisha b. Abuyah says, "One who has good deeds to his credit and has studied the Torah a great deal—to what is he to be likened?

B. "To someone who builds first with stones and then with bricks. Even though a great flood of water comes and washes against the foundations, the water does not blot them out of their place.

C. "One who has no good deeds to his credit but has studied the Torah—to what is he to be likened?

D. "To someone who builds first with bricks and then with stones. Even if only a little water comes and washes against the foundations, it forthwith overturns them." (Neusner 1997, p. 121)

This rendition of the parable compares two types of individuals based on their actions and the Torah study. The first person, with both good deeds and extensive Torah knowledge, is likened to a builder who constructs with stones at the foundation and bricks at the top. Even when confronted with a great flood, the solid foundation remains intact, symbolizing that the combination of good deeds and Torah learning will withstand God's judgment. In contrast, the second person lacks good deeds but has studied the Torah, resembling a builder who constructs with bricks at the foundation and stones at the top. Even a small amount of water is enough to overturn the weak foundation, illustrating that mere knowledge acquisition without good deeds cannot pass God's judgment.

### 3.2. Two Forms of Piety

Despite *Avot D'Rabbi Nathan* being a collection of somewhat disconnected epigrams and maxims providing advice on various aspects of Israelites' daily life, its two primary themes or emphases can be recognized as the study of the Torah and the performance of good works. (Goldin 1945, p. 97) Notably, there are two versions of *Avot D'Rabbi Nathan*: one found as a minor tractate appended to the Talmud and another discovered by Solomon Schechter in the nineteenth century and published in 1887. (Goldin 1945, p. 97). According to Judah Goldin, each version reflects one of these two emphases.

Theme or emphasis of version I [i.e., the Talmudic version] of *Abot de Rabbi Nathan* is the study of the Torah. Again and again throughout the text our version [i.e., the Talmudic version] goes out of its way to underscore the importance of the Torah study. And this emphasis is so strong as to leave the impression that version I is primarily concerned with the study of the Torah, while version II [the Schechter version] would underline "good works" or "good deeds". (Goldin 1945, pp. 98–99)

Nevertheless, as far as the parable is concerned, the form of piety that engages both the study of the Torah and good works is the one that is taught and encouraged, while the one focused solely on study is admonished. Another rabbinic text, the sixth-century *mishnaic* tractate *Kiddushin* also says: "Study is greater, but not as an independent value; rather, it is greater as study leads to action" (Sefaria 2022b, 40b:8). A broader cosmic perspective is also presented in *Avot D'Rabbi Nathan*, attributed to Shimon the Righteous, saying that "On three things does the world stand, on the Torah, and on the Temple service, and on deeds of loving kindness." (Neusner 1997, p. 33).

The Temple service, of course, has been suspended since the destruction of the Temple in 70 CE, but the calling remains in the Torah, and it is accentuated in שְׁמַע *Shema* (Hear), which is a central aspect of Judaism, consisting of three scriptural passages (Deuteronomy 6:4–9, 11:13–21; Numbers 15:37–41) coupled with prayers (Britannica 2023). It is integral to morning and evening services and is named after its opening word: "Hear, O Israel: The Lord is our God, the Lord alone. You shall love the Lord your God with all your heart and with all your soul and with all your might". (Deut. 6:4). It admonishes Israelites not to serve or worship other gods (Deut. 11:16) and instructs them to keep their body and mind clean (Num. 19:20). What the rabbinic literature (*Avot D'Rabbi Nathan* in particular) emphasizes is that, in addition to all these, one should do good works for fellow human beings ("deeds of loving kindness").

### 3.3. Two Worlds

The terms "עוֹלָם הזה *olamhazeh* (this world)" and "עוֹלָם הַבָּא *olamhaba* (the world to come)" are frequently employed in rabbinic literature to describe the individual's journey from one realm to the next. Neusner explains:

> If we were to ask the authorship of *Avot* to spell out their teleology, they would draw our attention to the numerous sayings about this life's being a time of preparation for the life of the world to come, on the one side, and to judgment and eternal life, on the other. The focus is on the individual and how he or she lives in this world and prepares for the next. The category is the individual, and, commonly in the two documents before us when we speak of the individual, we also tend to find the language of "this world" and "the world to come", *olamhazeh*, *olamhaba*. (Neusner 1997, p. xlvii)

In the journey between *olamhazeh* and *olamhaba*, the latter is the destination, the desired and sought-after place, while the former serves as the temporary passage and a ground for preparation. As *Pirkei Avot* puts it: "The world is like a vestibule before the world-to-be; prepare yourself in the vestibule, so that you may enter the banqueting-hall."(Sefaria 2022e, 4:16). Building upon this teaching, the eighteenth-century *Mesilat Yesharim* expands further, stating: "But the path to arrive at the 'desired haven' (Ps. 107:30) of ours is this world. This is what our sages of blessed memory said: 'this world is like a corridor before the world-to-come,'" (Sefaria 2022c, 1:3). and "The general principle of this matter: man was not created for his state in this world, but rather, for his state in the world-to-come. Only that his state in this world is a means towards his state in world-to-come, which is his ultimate purpose." (Sefaria 2022c, 1:15).

The following sayings from *Avot D'Rabbi Nathan* teach that a person who does not share part of what they earn with others in *olamhazeh* will not receive anything in *olamhaba*, and they allude to the framework of *olamhazeh* and *olamhaba* being implied in Ezekiel 2:10. Moreover, they highlight the existence of a trade-off between the pleasures experienced in *olamhazeh* and *olamhaba*. These three points collectively provide insights into the interconnection between actions in this world and their consequences in the world-to-come, as well as the dynamics of pleasure and rewards between the two realms.

> Para. A, Section VI, Chapter XII:

> A. Just as a person does not share in the wage of his fellow in this world, so he does not share in the wage of his fellow in the world to come … (Neusner 1997, p. 78)

> Paras. A–D, Section II, Chapter XXV:

> A. And there was written therein lamentations and jubilant sound and woe (Ez. 2:10):

> B. Lamentations refers to the penalty inflicted on the wicked in this world, …

> C. … and jubilant sound and woe refers to the reward of the righteous in the world to come, …

> D. … and woe: refers to the punishment that is coming to the wicked in the world to come, …" (Neusner 1997, p. 126)

> Paras. A–B, Chapter XXVIII:

> A. R. Judah the Patriarch says, 'From whoever is glad to get the pleasures of this world are held back the pleasures of the world to come.

> B. 'And to whoever is not glad to get the pleasures of this world are given the pleasures of the world to come'." (Neusner 1997, p. 138)

There is an emphasis that there will be no opportunity to do good works in *olamhaba*, even if one wishes to. The sixth-century *Avodah Zarah* says:

> What is the meaning of that which is written: 'You shall therefore keep the commandment, and the statutes, and the ordinances, which I command you this day,

to do them' (Deuteronomy 7:11)? This verse teaches that today, in this world, is the time to do them, but tomorrow, in the world-to-come, is not the time to do them. Furthermore, today is the time to do them, but today is not the time to receive one's reward, which is granted in world-to-come. (Sefaria 2022a, 3a:11)

The *olamhazeh-olamhaba* regime does not appear to be eschatological in the sense of one world ending and another beginning. Instead, as Neusner observes, it relates to "the individual and how he or she lives in this world and prepares for the next". The two worlds exist concurrently, and a person transitions from one to the other after completing life in this one, akin to moving from one space to another. This perspective is notably emphasized in the twelfth-century *Mishneh the Torah, Repentance*:

The reason why the sages styled it the world-to-come is not because it is not now in existence and will only come into being when this world shall have passed away. That is not so. The world-to-come now exists, as it is said, 'which Thou hast treasured up (for them that fear Thee), which Thou hast wrought (for them that trust in Thee before the children of men") (Ps. 31:19–20). It is called the world-to-come, only because human beings will enter into it at a time subsequent to the life of the present world in which we now exist with body and soul, and this existence comes first. (Sefaria 2022d, 8:8)

## 4. The Parable in the Sermon on the Mount of the Gospels

On a mountain in Matthew and a level place in Luke, Jesus delivers His famous sermon to disciples and a multitude. The sermon concludes with the parable of the two builders, stressing the importance of practicing His teachings, not just hearing them.

### 4.1. Imagery

The Sermon on the Mount in Matthew 7:24–27 (NRSV) states:

7:24 Everyone then who hears these words of mine and acts on them will be like a wise man who built his house on rock.

7:25 The rain fell, the floods came, and the winds blew and beat on that house, but it did not fall, because it had been founded on rock.

7:26 And everyone who hears these words of mine and does not act on them will be like a foolish man who built his house on sand.

7:27 The rain fell, and the floods came, and the winds blew and beat against that house, and it fell—and great was its fall!

The Sermon on the Plain in Luke 6:47-49 (NRSV) records:

6:47 "I will show you what someone is like who comes to me, hears my words, and acts on them.

6:48 That one is like a man building a house, who dug deeply and laid the foundation on rock; when a flood arose, the river burst against that house but could not shake it, because it had been well built.

6:49 But the one who hears and does not act is like a man who built a house on the ground without a foundation. When the river burst against it, immediately it fell, and great was the ruin of that house.

The parable portrays a wise man who builds his house on rock (Matthew), even digging deep to reach the rock if necessary (Luke). In contrast, a foolish man builds his house on sand (Matthew) or on the surface of the land (Luke). When nature tests the houses, with rain, flood, and wind (flood only in Luke), the house built by the wise stands, whereas the one built by the foolish completely collapses. The wise man symbolizes those who not only hear the words of Jesus but also act upon them, while the foolish man represents those who only listen but take no action. When the end time and judgment arrive, those in the latter category will face great suffering.

### 4.2. Two Forms of Piety

Regarding the words of Jesus referred to in the parable, they are understood to be His teachings in the Sermon on the Mount, rather than His teachings in the Gospels generally. Strecker highlights the demonstrative pronoun τούτους ('these') in 7:28 (and likely in 7:24 as well), which clearly points back to the words of the Preacher on the mount (Strecker 1988, p. 172). Hearing His words means receiving His teachings on the Beatitudes, love for enemies, almsgiving, prayer, fasting, the Golden Rule, hypocrisy of judging, wealth, anxiety, prayer, and the closing admonitions and parables.

The crowds were astounded at His authoritative teaching, distinguishing Him from their scribes (Matt 7:28–29). As Strecker notes, Jesus' teachings rest squarely on His own authority as the eschatological Lord:

> The Sermon on the Mount cannot be properly understood apart from the person of Jesus. Its interpretation cannot ignore the fact that Jesus as the teacher of the Sermon on the Mount is at the same time the eschatological Lord and Son of God, the revealer of God's will, as he is presented by the prologue of Matthew's Gospel (1: 18ff) and also by his shout of joy and redemption (11:25–30). This teacher is not simply comparable to a Jewish scribe. His teaching contains an eschatological call to decision, for it is expressed with divine ἐξουσία ("authority" 7:29). It points ahead to the eschaton and through teaching makes it present. Therefore, the ethical admonition of Jesus in the Sermon on the Mount is an eschatological demand. (Strecker 1988, p. 26)

Listening to and understanding the teachings of Jesus is important but putting them into action is equally essential. Acting on His words involves embodying the spirit of the Beatitudes, such as being humble, compassionate, meek, seeking righteousness, showing mercy, being pure in heart, promoting peace, and practicing forgiveness. It also entails giving to those in need, loving enemies, being genuine and honest, avoiding hypocrisy, trusting in God, refraining from judging others, and adhering to the principle of "in everything do to others as you would have them do to you" (Matt 7:12).

### 4.3. Two Worlds

The term *kosmos* κόσμος (the world) appears once in the Sermon, in Matt 5:14–16:

> 5:14 You are the light of the world. A city built on a hill cannot be hid.

> 5:15 People do not light a lamp and put it under the bushel basket; rather, they put it on the lampstand, and it gives light to all in the house.

> 5:16 In the same way, let your light shine before others, so that they may see your good works and give glory to your Father in heaven.

Here, *kala erga* καλὰ ἔργα (good works) refer to putting the teachings of Jesus into action, and the purpose of these good works is to serve as "the light of the world", illuminating the way for others so that they may witness and glorify God the Father in heaven.

In contrast to *kosmos*, *basileia ton ouranon* βασιλεία τῶν οὐρανῶν (kingdom of heaven) appears about ten times in the Sermon, warning that not all who call Jesus "Lord, Lord" will enter the kingdom of heaven (Matt 7:21–23); emphasizing that those who follow and teach the commandments will be esteemed highly in the kingdom of heaven (5:19–20); providing instructions on praying for and seeking the coming of the kingdom (6:10, 33); and assuring through the Beatitudes that the poor in spirit and persecuted will inherit the kingdom of heaven (5:3, 10, 6:33). The eschatological nature is grounded in Jesus being the eschatological Lord, the general eschatological outlook of the Sermon, and is explicitly revealed in verses like "Blessed are the poor in spirit, for theirs is the kingdom of heaven" (5:3) and "Blessed are those who are persecuted for righteousness' sake, for theirs is the kingdom of heaven" (Matt 5:10). Entering the kingdom of heaven and receiving rewards hinge on one's willingness to carry out the good works commanded by Jesus.

The kingdom of heaven does not primarily carry an anthropological or ethical connotation; instead, it represents a cosmic domain transcending humanity, and its eschatological fulfillment will encompass both heaven and earth (Strecker 1988, p. 114). As evident from the cited passages, the kingdom represents a future realm that is hoped for and anticipated. This world, on the other hand, serves as the stage for such hope and anticipation, and it is not prominently highlighted. When mentioned, its primary meaning is to allow for a time and stage for evangelization and salvation, rather than for individuals doing the right thing in this world so as to be compensated in the next.

## 5. Comparing the Versions

In this section, we will compare the three versions of the parable and determine which —Builders/Rabbinic or Builders/Gospels—bears a closer resemblance to Builders/YSL.

### 5.1. Imagery

Both Builders/YSL and Builders/Rabbinic center their imagery on a house, and the choices are whether the house is to be fastened or not to the ground (Builders/YSL), or whether it is to be built with stones down first or bricks down first (Builders/Rabbinic). In contrast, Builders/Gospels focuses on the terrain, and the choices are between a rocky site or a sandy site.

### 5.2. Piety

Each rendition of the parable underscores the significance of virtuous deeds, highlighting the notion that piety devoid of corresponding actions remains inherently incomplete. Beyond this commonality, notable parallels emerge between the teachings of rabbinic tradition and the spiritual philosophy of YSL. In the realm of rabbinic piety, the meticulous study of Torah, with the *Shema* prayer serving as its focal point, accentuates the divine unity, cautions against the worship of other deities, and champions both spiritual and physical purity. Similarly, YSL underscores earnest devotion to a singular God, warns against demonic worship, emphasizes obedience to divine directives, and advocates for the preservation of mental and physical integrity. In stark contrast, the Builders/Gospels pivots around the centrality of Jesus as the Lord and Son of God, urging adherence not only to His teachings but also their practical implementation. The teachings within Builders/Rabbinic and Builders/YSL exhibit a distinctly human presentation, deriving authority solely from the role of a rabbi or teacher. Conversely, the Gospel iteration is positioned as a divine teaching, carrying the authoritative weight of the Son of God.

### 5.3. Two Worlds

Builders/YSL and Builders/Rabbinic both underscore that present actions shape the future, emphasizing preparation in this world and the absence of opportunities in the world-to-come. These two traditions exhibit strong textual alignments, portraying the concurrency of the two worlds that coexist. Conversely, Builders/Gospels introduces "*kosmos*" and the "kingdom of heaven", which are not concurrent but consecutive, emphasizing cosmic transformation and divine fulfillment beyond earthly life. Hence, Builders/YSL resonates more with Builders/Rabbinic, focusing on molding the future through present actions, while Builders/Gospels presents a vision of eschatological consecutiveness, portraying the present world as a time or dispensation for spreading the gospel.

A striking terminological parallel exists between the paired terms "*citianxia-bitianxia*" in YSL and the paired terms "*olamhazeh-olamhaba*" in rabbinic literature. Notably evident in rabbinic literature, the term "*olamhaba*", though absent from the Hebrew Bible or Second Temple Hebrew writings, appears frequently, exceeding two thousand times, often accompanied by "*olamhazeh*". (Weiss 2017, p. 91). A corresponding pattern exists in YSL, where the terms "*citianxia*" and "*bitianxia*", along with their variations, feature 58 times within 60 lines of text. This significant terminological symmetry underscores the profound

affinity shared between YSL and rabbinic literature in encapsulating the concept of two interconnected worlds.

The comparison is summarized in the Table 1 below:

**Table 1.** Comparison of Parable of Wise and Foolish Builders in YSL, Rabbinic Literature, and the Gospels.

| | Builders/YSL | Builders/Rabbinic | Builders/Gospels |
|---|---|---|---|
| **Imagery** | Centers on the house | Centers on the house | Centers on the terrain |
| -   Choices | -   anchored or not to the ground. | -   stones down first or bricks down first. | -   rocky site or sandy site |
| **Two Forms of Piety** | Centers on God | Centers on God | Centers on Jesus, the Lord and Son of God. |
| -   First form | -   loving, worshipping, and obeying the only one God, adhering solely to His commands, maintaining cleanliness in mind and body, rejecting the worship of demons and purging one's heart of malice, venom, revenge, or jealousy. | -   studying the Torah, the core of which is the *Shema* prayer calling for the loving, worshipping, and serving God, rejecting other gods, keeping the body and mind clean. | -   hearing the word of Jesus, the Lord and Son of God, which covers the Beatitudes, trust in God, the Golden Rule, honesty, kindness, love for people (even enemies), avoidance of judging and hypocrisy. |
| -   Second form | -   doing all the above plus doing good works, which mainly means almsgiving. | -   doing all the above plus doing good works, which mainly means almsgiving. | -   hearing and doing what Jesus, the Lord and Son of God, has taught. |
| -   Divine authority Claimed | -   no | -   no | -   yes |
| **Two Worlds** | Two realms of human existence | Two realms of human existence | Two domains in God's salvation plan |
| -   Relationship | -   cosmologically concurrent, with this life in this world being the opportunity to earn merits for reward and a better life in the next. | -   cosmologically concurrent, with this life in this world being the opportunity to earn merits for reward and a better life in the next. | -   eschatologically consecutive, with life in this world being the light to bear witness for Christ and to save people. |
| -   Textual alignments | -   "*citianxia-bitianxia*", featuring 58 occurrences within the 60 lines of text. | -   "*olamhazeh-olamhaba*", exceeding two thousand occurrences in rabbinic literature. | - |

To sum up, the findings assert that, in terms of metaphor and religious language, Builders/YSL and Builders/Rabbinic exhibit a closer affinity. They emphasize the house and two methods of construction, while Builders/Gospels centers around the site and two types of terrain. Concerning piety, both Builders/YSL and Builders/Rabbinic, as human teachings, exhort fellow humans to dedicate themselves to God, emphasizing the maintenance of spiritual, mental, and physical purity. In contrast, Builders/Gospels presents a divine set of teachings to be heard and implemented in daily life. Regarding the two worlds, Builders/YSL and Builders/Rabbinic depict them as concurrent, with divine judgments upon transition, while Builders/Gospels envisions eschatologically consecutive worlds,

with judgment at the end of time.[21] Notably, YSL's "*citianxia-bitianxia*" and the rabbinic "*olamhazeh-olamhaba*" are terminologies closely in parallel, and such an equivalent cannot be found in Builders/Gospels.

## 6. Conclusions

Traditionally, YSL has been designated as an "Aluoben document" in the *Jingjiao* corpus, a classification introduced by Saeki in 1937 (Saeki [1937] 1951, p. 113). This classification primarily relied on aligning YSL's dates (circa 641 CE) with the arrival of Aluoben in Chang'an (635 CE), as documented in the *Jingjiao Daqin Jingjiao Liuxing Zhongguo Bei* 大秦景教流行中國碑 [Monument Commemorating the Propagation of Daqin Jingjiao in China] ("the Monument") (Saeki [1937] 1951, pp. 114–5). The underlying assumption was that during the Tang Dynasty, only one history or tradition of Christianity existed in China, as narrated by the Monument, leading to the *a priori* position that all Christian relics or manuscripts from that period must be, without question, *Jingjiao*-related. While this association may seem reasonable for manuscripts sharing the same name as the Monument (i.e., "*Daqin Jingjiao* 大秦景教"), verification is essential for those, like YSL, that do not. The hypothesis of categorizing everything under *Jingjiao* needs scrutiny, and unfortunately, both Saeki's works and contemporary *Jingjiao* scholarship have shown a lack of initiative in conducting verifications through methods such as textual criticism.

In this study, we identified parallels between YSL and rabbinic literature, specifically in their use of the parable of the Wise and Foolish Builders and associated passages. Tentatively, we could infer that YSL, despite being a Christian document, incorporates distinctive elements from the Jewish tradition. Across publications since 2018, this author has demonstrated that YSL presents a unique set of divine designations, suggesting a potential connection to a distinct Christian community and tradition (Tam 2018a, 2018b, 2021b). Additionally, he has outlined its overarching theme, purpose, and structure (Tam 2022a), questioned its association with the Monument and Aluoben, cast doubt on Chang'an as the probable place of authorship, and proposed that western locales such as Gaochang (Turfan) in *Xiyu* (Western Region) are more likely settings for its creation (Tam 2021a, 2022b). These findings hint at the possibility that YSL was composed by a Christian community with a Jewish heritage in the seventh century, in *Xiyu*. Regardless, the document remains an open field for exploration and scholarly investigation, demanding rigorous exegesis and textual criticism.

**Funding:** This research received no external funding.

**Institutional Review Board Statement:** Not applicable.

**Informed Consent Statement:** Not applicable.

**Data Availability Statement:** Data are contained within the article.

**Conflicts of Interest:** The authors declare no conflict of interest.

## Notes

[1]   The term "*Yishen*" comprises two words, where "*yi*" means "one" and "*shen*" means "god". In *Jingjiao* scholarship, it is commonly rendered as "One God" or "one God". However, given that "*Yishen*" primarily functions as an appellation for God in YSL, essentially serving as a proper noun (refer to Tam 2018a, 2018b, 2021b), it is fundamentally synonymous with "God" (with a capital letter "G") in English. In the latter scenario, "God" is not prefixed with "One" or "one", unless the specific context necessitates it (as seen in Malachi 2:10, 1 Corinthians 8:6, etc.); a similar approach should be taken when translating "*Yishen*".

[2]   Haneda derived this date from lines 365–366: "自尔已來,弥師訶向天下見,也向五蔭身六百四十一年不過,已於一切處." Sun Jianqiang critiques this sentence as "ungrammatical and unintelligible," due to the placement of the term "*buguo* 不過 (not more than)" (Sun 2018, p. 140). The current author has illustrated, by citing comparable sentence structures from other Dunhuang manuscripts, that the sentence is grammatical and intelligible. (Tam 2022b, p. 27), and it means: "From then till now, since Mishihe [Christ] descended to the world and to the human body, it has not been more than 641 years, and it [the sacred transformation] is now everywhere." Separately, it should also be noted that if the author of YSL (and his community) was using the

Seleucid calendar, the date of YSL would be approximated to 635–637 CE (Takahashi, Hidemi. "Re: Yishenlun and year 641." Message to David Tam. 30 November 2017. E-mail).

3   The latest publication of the manuscript, with line numbers and in colour, is in (Tono 2020, pp. 23–53). A more accessible copy of the text, also with line numbers but in black and white, can be found at (Lin 2003, pp. 350–86).

4   P. Y. Saeki interpreted the phrases "*yu dier* 喻第二" at the bottom of line 60 as the subtitle for lines 1–60, the phrase "*yitian lun diyi* 一天論第一" at line 206 as the subtitle for lines 61–205, and the phrase "*Shizun bushi lun disan* 世尊布施論第三" at line 207 as the subtitle for lines 208–404. (Saeki [1937] 1951, pp. 8, 113) However, Saeki did not explain why, whereas the third "subtitle" appears at the beginning of its section, the first two appear at the end of theirs. He also did not demonstrate how these "subtitles" reflected the content of their respective parts or conveyed the overall structure of the document. Saeki's three-part division of YSL, however, has persisted in scholarship, with these issues remaining unresolved. This article suggests setting aside these problematic "subtitles" and aims instead to understand the document's structure by analyzing its content.

5   YSL uses the title "*Shizun* 世尊"as a combined designation for Jesus in passages where the Gospels variably use the epithets "Son of God," "Word/Logos," and "Lord" (Tam 2018a).

6   Huang Xianian identified 39 items (Huang 2000), and Nie Zhijun identified 54 (Nie 2010).

7   The relevancy of Dunhuang and Turfan manuscripts as reference materials in the interpretation of YSL is presented and argued in (Tam 2022b).

8   "*Shi*" 事 is translated as "love and serve", referencing "資於事父以事母,而愛同" ("as they serve their fathers, so they serve their mothers, and they love them equally") in *Xiaojing*《孝經》(The Classic of Filial Piety). The term "*Tianzun*天尊" corresponds to "*Yahweh*" or its substitute "*Adonai* (the Lord)" in the Jewish Bible. (See Tam 2018b, 2021b). (For other interpretations: (Saeki [1937] 1951, p. 185) "let him serve sole-heartedly this one God, the Lord of Heaven"; (Tang 2002, p. 166) "Serve only the One-God, the heavenly Lord"; (Aguilar Sanchez 2021, p. 170) "Serve only the One-God, the Heavenly Respected"; (Nicolini-Zani 2022, p. 241) "serve only the One God"; (Weng 1995, p. 126) "事,侍奉").

9   *Yiqu Yishen jinzhi* 一取一神進止: The terms *yiqu* 一取 and *jinzhi* 進止 appear quite often in Dunhuang manuscripts, in texts dealing with official or military affairs. In such narratives, "*yiqu*" (一取) conveys the concept of "exclusively taking orders or instructions from", while "*jinzhi*" (進止) pertains to "proceeding as per instructions". As such, *yiqu Yishen jinzhi* can be translated as "obey the commands of God only". Dunhuang references: (Pelliot 3213 in (Wang 1957, p. 26): "吳王致疾臨死之時,咐囑太子夫差:'汝後安國治人, 一取國相子胥之語.'"; Stein 2144 in (Wang 1957, p. 200): "衾虎昇帳而坐, 遂喚一官健只在面前, 再三處分:'公解探事,一取將軍處分,探得軍機,速便早迴,与公重賞.'"; (Pelliot 2942 in (Tang and Liu 1986, vol. 2, p. 631): "各牒所由, 准狀勘報,當日停務,勿遣東西,仍錄奏聞,伏待進止."; (Stein 6836 in (Wang 1957, p. 221): "開元皇帝好道,不敬釋門,遂命中使至玄都觀內宣進止,詔淨能."(For other interpretations: (Saeki [1937] 1951, p. 185) "Let him obey only what is commanded by this one God"; (Tang 2002, p. 166) "Seek the One-God. Do in such a manner"; (Aguilar Sanchez 2021, p. 170) "be the one who holds on to the instructions of the One-God"; (Nicolini-Zani 2022, p. 241) "and submit to the orders of the One God"; (Weng 1995, p. 126) "取,通; 進止, 舉止"; (Wu 2015, p. 98) "專心致意地尋求上帝的旨意.")

10  One of the first tasks in the study of "*Jingjiao* documents," or Dunhuang manuscripts in general, is to comb through the text to determine where sentences begin and end (adding punctuation), and whether characters are repeated or omitted due to copyist errors. Scholars' efforts in this regard can be found in (Saeki [1937] 1951, vol. IV, pp. 1–96; Weng 1995, pp. 107–49; Wang 2016, pp. 173–235, etc.), and most often their works (the edited texts) are simply shown, not explained. Here, we propose to delete "不是" from "不是此意知功德", and our explanation is that after deletion, the sentence"此意知功德,不是餘處功德"will be parallel with the subsequent sentence "此處功德,不是功德處", with the first part of the sentence stating the subject matter, and the second part refuting it. This rhetorical form is highly noticeable in the *citianxia-bitianxia* passage, especially in lines 130–140. More on this in the next footnote.

11  This sentence intends to contrast "*yizhi gongde* 意知功德" with "*yuchu gongde* 餘處功德". The term "*yuchu*" is used twice in YSL, first in lines 127–128: "如天下人,盡皆是母胎中所作,餘處不能作. (All people in the world are born from the mother's womb, for it is impossible elsewhere)". The term generally means "other", and in 127–128 it is used to contrast "the mother's womb", referring to places other than the mother's womb. In the current context, "*yuchu gongde* 餘處功德" means the good works other than "*ci yizhi gongde* 此意知功德", i.e., the good works of the mind (worshipping and self-cultivation). The contrast of the two would yield the translation "this is the good works of the mind, but not all the good works". As such, the first "不是" in "不是此意知功德" should be regarded an insertion error and should be deleted. This pattern of affirmation and negation is repeated in the following sentence, supporting our interpretation of this sentence. (For other interpretations: (Saeki [1937] 1951, p. 185): "Unless you understand the meaning of a meritorious deed in this sense, it is not a meritorious deed at all". Tang (2002, p. 166): "Ignorance of such significance leads to no merits and virtue"; (Aguilar Sanchez 2021, p. 170): "If you haven't got this mindset, acquire no merit and virtue"; (Nicolini-Zani 2022, p. 241): "If the meaning of this is not understood, then there will not be any meritorious actions"; (Weng 1995, p. 127): "'不是此意知', 不知此意, 即事一神, 禮拜一神. '功德不是', 不能算功德"; (Wu 2015, p. 98): "'此意', 此指'禮拜一神,一取一神進止'之真心實意. '餘處', 別處.").

12  (For other interpretations: (Saeki [1937] 1951, p. 185): "It may be a meritorious deed of other place (i.e., sect), but it cannot be a meritorious deed of this place (i.e., sect)"; (Tang 2002, p. 166): "Merits and virtues in other places are not the merits and virtues in this place"; (Aguilar Sanchez 2021, p. 170): "here and there, will not find the venue to acquire merit and virtue"; (Nicolini-Zani 2022, p. 241): "Neither this nor other worlds will be places [in which to foster] meritorious actions".).

[13]    (For other interpretations: (Saeki [1937] 1951, pp. 185–86): "To do a meritorious deed, for instance, is like a man building a house. First of all, before he builds it, he makes the piles for the house. He must, above all, fix these piles firmly and strongly". (Tang 2002, p. 166): "Where merits and virtues are like where one builds a house. The foundation has to be laid beforehand and be solidly laid"; (Aguilar Sanchez 2021, p. 170): "For instance, when someone who wants to build a house, must prepare for the foundation of the house, a solid foundation"; (Nicolini-Zani 2022, p. 241): "To make use of a simile, someone who is building a house first has to lay the foundations"; (Weng 1995, p. 126): "'作基', 打基礎"; (Wu 2015, p. 99): " '基腳', 牆根；牆腳.").

[14]    "*Xiuxing* 脩行" typically means "self-cultivation", a discipline often associated with study of classics and scriptures. Dunhuang References: Pelliot 3432 in (Tang and Liu 1986, vol. 3, p. 3): "修行道地經陸卷,生經等伍卷."; and Stein 4571 in (Wang 1957, p. 520): "所以如來說此經,總教平穩行心識. 經文正引好修行,只是徒心發赤誠." "*Jujie* 具戒" is translated as "observe the rules". Dunhuang references: Pelliot 4660 in (Tang and Liu 1986, vol. 5, p. 141): "一從披削,守戒修禪."; and Pelliot 4660 in (Wang 1957, p. 147): "精持戒律,白日無虧.". (For other interpretations: (Saeki [1937] 1951, p. 186): "… to observe all the rules and precepts of life set up (by God) and to prepare himself perfectly for the deed"; (Tang 2002, p. 166) "… one has to cultivate oneself first and be fully vigilant"; (Aguilar Sanchez 2021, p. 170): "… must try first to sanctify oneself into a perfectly prepared wholeness"; (Nicolini-Zani 2022, p. 241) "First of all, one must correct one's own actions and be perfectly vigilant"; (Wu 2015, p. 99): " '修行', 修養德行. '具戒', 具足圓滿天尊之戒." (Wang 2016, p. 222): "'修行',佛教術語,四法之一,如理修習作行也, 通於身語意之三業. '具戒備足',佐伯本校作 '戒備具足',不確. 此處文意可通.).

[15]    "別作" is translated as "also do" with reference to Pelliot 2187 in (Wang 1957, p. 347): "喚風伯雨師作一營,呼行病鬼王別作一隊." (For other interpretations: (Saeki [1937] 1951, p. 186): "… and then they will do more meritorious deeds than ever"; (Tang 2002, p. 166): "Then one can do other things of merit and virtue"; (Aguilar Sanchez 2021, p. 170): "… then it is not necessary to push oneself for merit and virtue"; (Nicolini-Zani 2022, p. 241): "Only in this way can they then perform meritorious actions"; (Weng 1995, p. 127) "'更別作', 再別作"; (Wu 2015, p. 99): " '更別', 再另外行、'更', 再、'別', 另外.").

[16]    "*Xuanchui* 懸吹": In Dunhuang manuscripts, the *xuanchui* 懸吹 or *xuanfeng* 懸風 cannot be found, but terms such as "*xuani* 懸泉 (hanging spring or water)", "*xuanjian* 懸劍 (hanging sword)", "*xuanya* 懸崖 hanging cliff", "*xhuanshe* 懸蛇 (hanging snake)" are common, suggesting that the *xuanchui* 懸吹 or *xuanfeng* 懸風 to mean a wind hanging from the sky, i.e., a swirling wind like a dust devil or even tornado.

[17]    "處分" is translated as "order" or "will" with reference to Pelliot 2962 in Stein 133 in (Wang 1957, p. 155): "女生外向,千里隨夫,今日屬配郎君,好惡聽從處分." Stein 2073 in (Wang 1957, p. 169): "臣奉大王處分,遍歷山川,搜尋精靈狐魅,並不見一人." (For other interpretations: (Saeki [1937] 1951, pp. 183–84) "Be sure not to act contrary to what is commanded by the one God"; (Tang 2002, p. 165): "The punishment from the One-God cannot be run against"; (Aguilar Sanchez 2021, p. 168): "The laws/instructions of the One-God must be obeyed". (Nicolini-Zani 2022, p. 240): "The One God has disposed that he should not be disobeyed, …"; (Wu 2015, p. 99): "'處分', 吩咐之意.").

[18]    "*Faxin* 發心" is translated as "the resolve" with reference to Stein 2073 in (Wang 1957, p. 188): "眾生發心修道,先須讀誦經文."; Pelliot 3721 in (Tang and Liu 1986, vol. 1, p. 82): "辛酉開元九年,僧處該與鄉人百姓馬恩忠等發心造南大像."; Stein 4571 in (Wang 1957, p. 524): "時阿難既聞仏語,途即發心離諦,受已歸依."; Pelliot 2324 in (Wang 1957, p. 401): "我今發心求剃度,師兄緣甚暑艱難？" (For other interpretations: (Saeki [1937] 1951, p. 184): " … one should be generous and magnanimous"; (Tang 2002, p. 165): "The heart of expression should be wide open"; (Aguilar Sanchez 2021, p. 170) "The charitable heart/mind must be kept enormous/big, …"; (Nicolini-Zani 2022, p. 240): "A generous heart is one that is wide and large"; (Weng 1995, p. 125): "'發心', 發意動念."; (Wu 2015, p. 99): " '發心', 發出憐憫慈善之心." (Wang 2016, p. 221): " '發心',佛教術語,發菩提心也,願求無上菩提之心也.").

[19]    "*Hezou* 合作" is translated as "appropriate (or suitable) to be done" in reference to Pelliot 2144 in (Wang 1957, p. 206): "夜來三更奉天苻牒下,將軍合作陰司之主." (For other interpretations: (Saeki [1937] 1951, p. 183): "… if there be any who seek (to enter) that world they must do so whilst in this world. If he cannot do (good deeds) whilst in this world, …"; (Tang 2002, p. 165): "It is like what will be needed in the next world. They must be jointly prepared in this world. If they were not jointly made in this world, …"; (Aguilar Sanchez 2021, p. 168): "It is like what will be needed in the other world, all must collaborate, if they do not work together, …"; (Nicolini-Zani 2022, p. 239): "This world cooperates with what will be in the other world. If this world were not to collaborate, …").

[20]    For the sentence "一神自聖化,神自聖化", we take the understanding that the second "神自聖化" is a repetition due to copyist error.

[21]    In lines 98–101, YSL articulates a perspective on the eschaton, as expressed in the passage: "若天地滅時,劫更生時,魂魄還歸五蔭身來 (If when tiandi are destroyed and a new eon is born, the soul returns to the wuyin body, …)." It is crucial to observe that YSL employs the term "*tiandi* 天地 (cosmos)" (not "*tianxia*") to denote the entity subjected to processes of destruction and renewal.

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
