# Peer review of "The Parable of Wise and Foolish Builders in Yishen Lun and Rabbinic Literature"

_religions, doi:10.3390/rel15010107_

Round 1

Reviewer 1 Report

Comments and Suggestions for Authors

It would be worthwhile to expand the arguments in §2 re the interpretation of YSL rather than just noting catchwords/phrases from Saeki and Nicolini-Zani, especially since there is an overall lack of consensus.

References occasionally need to be supplied, e.g. the Schechter discovery of Avot d'Rabbi Nathan published in 1887 especially since references are made to it.

Comments on the Quality of English Language

English expression is satisfactory, however it would be preferable for the author to spell out plene words such as it's (= it is) and there's (= there is) for a printed publication.

Minor slips: p.9 last line: alludes read allude; p.10 l.1 highlights read highlight

Author Response

It would be worthwhile to expand the arguments in §2 re the interpretation of YSL rather than just noting catchwords/phrases from Saeki and Nicolini-Zani, especially since there is an overall lack of consensus.

As indicated in the first paragraph of §2, this author presents his arguments and rationale for his interpretations of YSL in the footnotes. 

References occasionally need to be supplied, e.g. the Schechter discovery of Avot d'Rabbi Nathan published in 1887 especially since references are made to it.

Two references have been added in this regard. A general checking has been done to beef up the references throughout.

English expression is satisfactory, however it would be preferable for the author to spell out plene words such as it's (= it is) and there's (= there is) for a printed publication.

These corrections have been made. Thank you.

Minor slips: p.9 last line: alludes read allude; p.10 l.1 highlights read highlight

Correction made. Thank you.

Reviewer 2 Report

Comments and Suggestions for Authors

This is an exceptionally well researched and well written article which will make a good contribution to comparative scholarship regarding eastern Christianity.  The author shows an excellent knowledge of a several languages and manuscript traditions.  I found a three-way comparison to be particularly enlightening.  I am not entirely convinced by the importance of the differences between YSL and Rabbinic materials on the one hand, and the Gospel materials on the other, but nonetheless this is an excellent article which argues cogently for its point of view.  This article represents a high standard of scholarship.

Author Response

Thank you for your kind and encouraging comments.

In terms of the importance of the similarities/differences shown in this three-way comparison, I have beefed up the conclusion to say that “Tentatively, we could infer that YSL, despite being a Christian document, incorporates distinctive elements from the Jewish tradition,” and “Regardless, the document remains an open field for exploration and scholarly investigation, demanding rigorous exegesis and textual criticism.”

Reviewer 3 Report

Comments and Suggestions for Authors

At pp. 12 and 13, the world "kosmou" shall be changed into "kosmos" (x 3)

Comments on the Quality of English Language

Good

Author Response

Thank you. Corrections made.